# The Hidden Relationship between Intestinal Microbiota and Immunological Modifications in Preeclampsia Pathogenesis

**DOI:** 10.3390/ijms251810099

**Published:** 2024-09-20

**Authors:** Enrica Zambella, Beatrice Peruffo, Alice Guarano, Annalisa Inversetti, Nicoletta Di Simone

**Affiliations:** 1Department of Biomedical Sciences, Humanitas University, Via Rita Levi Montalcini 4, Pieve Emanuele, 20072 Milan, Italy; enrica.zambella@st.hunimed.eu (E.Z.); beatrice.peruffo@st.hunimed.eu (B.P.); annalisa.inversetti@hunimed.eu (A.I.); 2Humanitas San Pio X, Via Francesco Nava 31, 20159 Milan, Italy; alice.guarano@sanpiox.humanitas.it; 3IRCCS Humanitas Research Hospital, Via Manzoni 56, Rozzano, 20089 Milan, Italy

**Keywords:** preeclampsia, gestational hypertension, hypertensive pregnancy disorder, gut microbiota, gut metabolome

## Abstract

Preeclampsia is a multifactorial gestational syndrome characterized by increased blood pressure during pregnancy associated with multiorgan involvement. The impact of this disease on maternal and neonatal health is significant, as it can lead to various fetal comorbidities and contribute to the development of maternal comorbidities later in life. Consistent evidence has shown that the microbiota acts as a regulator of the immune system, and it may, therefore, influence the development of preeclampsia by modulating immune factors. This narrative review aims to investigate the role of the immune system in the pathogenesis of preeclampsia and to summarize the most recent literature on the possible link between preeclampsia and alterations in the intestinal microbiota. To this end, we conducted a literature search, aiming to perform a narrative review, on PubMed and Embase from January 1990 to March 2024, focusing on the latest studies that highlight the main differences in microbial composition between patients with and without preeclampsia, as well as the effects of microbial metabolites on the immune system. From the review of 28 studies assessing the intestinal microbiota in preeclamptic women, preeclampsia could be associated with a state of dysbiosis. Moreover, these patients showed higher plasmatic levels of endotoxin, pro-inflammatory cytokines, and T helper 17 cells; however, the findings on specific microbes and metabolites that could cause immune imbalances in preeclampsia are still preliminary.

## 1. Introduction

Hypertensive disorders of pregnancy (HDPs) represent an important topic in the obstetrical field. HDPs affect a variable number of pregnancies worldwide, with an incidence range that goes from 3% to 8–10% [1]. These conditions lead to a higher risk of adverse pregnancy outcomes like pre-term birth, intrauterine growth restriction (IUGR) [2], defined as estimated fetal weight (EFW)/fetal abdominal circumference (AC) below the 3rd percentile or as a drop in AC or EFW of >2 quartiles or >50 percentiles and abnormal Doppler velocimetry of the uteroplacental and fetoplacental circulations [3], and lastly maternal and/or perinatal death [1,4]. Indeed, HDPs still represent the sixth cause of maternal death in the United States (6.3% in the period 2017–2019) [5] and the fifth in Italy [6]. In the group of HDPs, preeclampsia (PE) is the most severe disease, characterized by de novo hypertension after 20 weeks’ gestation (systolic blood pressure (SBP) of ≥140 mmHg and/or diastolic (DBP) of ≥90 mmHg on two measurements executed at least 4 h apart, or SBP of ≥160 mmHg and/or DBP of ≥100 mmHg collected on a single measurement) and proteinuria (>300 mg/24 h or protein/creatinine ratio of ≥0.3 mg/dL). In the absence of proteinuria, at least one of these new-onset signs or symptoms should be identified in addition to hypertension: thrombocytopenia, liver dysfunction, renal insufficiency, pulmonary edema, and headache unresponsive to medication and not justified by alternative diagnoses or visual disturbances [1,7]. The International Society for the Study of Hypertension in Pregnancy (ISSHP) also includes uteroplacental dysfunction (e.g., placental abruption, fetal growth restriction defined as an estimated fetal weight <10th percentile, abnormal umbilical artery Doppler, or intrauterine fetal death) as a diagnostic criterion for PE [8]. Preeclampsia represents the second cause of maternal death after postpartum hemorrhage in developing countries [9] and, according to Ending Preventable Maternal Mortality (EPMM) strategies stated by the WHO, it is one of the highly preventable causes of obstetric adverse outcomes [10]. For this reason, extending our knowledge of the possible risk factors and developing a proper screening strategy is essential to identify and treat this condition in order to prevent most related deaths [11]. Moreover, over the past decade, PE has been associated with a higher occurrence of chronic diseases later on in both the mother’s and offspring’s lives [12,13]. There is increasing evidence about the possible role of altered immune regulation in preeclampsia, related to defective maternal–fetal immune tolerance and higher prevalence in patients affected by autoimmune diseases [14]. Hence, more systemic and complex mechanisms involved in preeclampsia pathogenesis need to be studied.

The gut microbiota is a complex association of microorganisms’ ecosystems, including bacteria, fungi, archaea, viruses, and protozoa, able to resist unfavorable gastrointestinal conditions in a strict symbiotic relationship with the host [15]. These microorganisms and their collective genomes (microbiome) have several essential metabolic functions (metabolome); therefore, the gut microbiota is also defined as the postnatally acquired “metabolic organ” [16]. The gut microbiota has an important influence on many developmental processes and functions of the host, in particular on the immune system. It is well known that the microbiota enhances the integrity of epithelial cells as a barrier against pathogens, and it is also involved in priming immune cells through the production of immunomodulatory metabolites [17,18,19]. In this regard, short-chain fatty acids (SCFAs) are metabolites produced by gut bacteria during fermentation of dietary fiber, and it has been demonstrated that they can influence the balance of pro-inflammatory and anti-inflammatory cytokines and can have a potential role in Treg cell development in offspring [20]. Moreover, considering that microbiota-derived metabolites, like lipopolysaccharide (LPS) and peptidoglycans, can cross the syncytiotrophoblast barriers, the development of immune cells starts in the womb, and this process can be influenced by the maternal microbiota [21]. In a study using 16S rRNA sequencing on human fetal meconium at 18–23 weeks of gestation, the presence of *Micrococcaceae* was linked to the development of specific intestinal memory T cells and processes like Toll-like receptor signaling, phagolysosome function, immune cell chemoattraction, and macrophage inhibition [22]. Similarly, another study demonstrated the presence of bacterial cocci within the fetal gut lumen at 14 weeks of gestation, able to elicit a memory response by the mesenteric lymph node T cells [23]. Structural changes in commensal microbial communities lead to dysbiosis, recently associated with several systemic inflammatory syndromes, such as inflammatory bowel diseases (IBDs) or asthma, and chronic diseases like hypertension, obesity, diabetes mellitus and metabolic syndrome [24]. These chronic diseases are well-known risk factors for PE development, and they also account for many of the long-term outcomes of PE in women’s lives [25]. In early pregnant obese and overweight women, specific gut microbiome compositions, such as higher levels of *Collinsella* and *Coprococcus*, have been linked to increased insulin resistance and abnormal lipid metabolism [26].

In this delicate balance, the microbiota may play a role in the pathogenesis of immune diseases, since a dysregulation occurring at any level of this complex process could potentially cause an immunological disorder. The impact of microbiota on the maternal immune system [26,27] as well as the interaction between intestinal dysbiosis and metabolic diseases leads to the hypothesis of the key role of microbiota in PE pathogenesis [28]. The principal aim of this narrative review is to describe the potential association between dysbiosis and preeclampsia compared to controls, the role of metabolomic analyses, and the clinical impact of microbial dysbiosis on maternal health.

## 2. Methods

We conducted a literature search from January 1990 to March 2024 across two electronic databases: PubMed and Embase. We used various combinations of the following keywords to analyze relevant articles: “gut microbiota” or “intestinal microbiota” or “gut metabolome” or “intestinal metabolome” and “preeclampsia” or “gestational hypertension” or “hypertensive pregnancy disorders”. We included randomized controlled trials, cohort studies, case–control studies, and cross-sectional studies in our review. Additionally, reference lists of retrieved reports, review articles, and meta-analyses were evaluated to identify potential additional studies. We excluded case reports, letters and non-English language publications. Two authors (E.Z. and B.P.) independently conducted electronic searches and analyzed a total of 264 studies.

## 3. Preeclampsia and Its Pathogenesis

The primary mechanisms behind PE are still partially unknown, even if some steps towards understanding its pathophysiology have been made in the last decade [29,30,31].

Placentation has a key role: cytotrophoblast cells invade the uterine myometrium (invasive phenotype) and shape efficient vascular anastomoses between the spiral arteries to support the placental circulation [32,33]. It has been demonstrated that in HDP/PE disease, cytotrophoblasts do not acquire the physiological invasive phenotype due to trophoblast immaturity, leading to an inadequate vascular invasion and narrow blood vessels. Therefore, the development of abnormal vessels results in endothelial damage and leads to insufficient oxygen transfer, placental ischemia, and consequent increased reactive oxygen species (ROS) production [34]. These alterations cause an imbalance in the immune tolerance mechanisms, causing the secretion of inflammatory molecules and an alteration between angiogenic and anti-angiogenic factor proportions. The increase in pro-inflammatory and anti-angiogenic proteins in maternal blood circulation represents the basis of the clinical syndrome, with its typical systemic signs and symptoms [35].

PE is commonly classified into two different subtypes according to a worldwide consensus, which differ in incidence, pathogenesis, and maternal–fetal outcomes: early-onset preeclampsia (EOPE), before 34 weeks of gestation, which accounts for a minority of all cases (5–20%) and late-onset preeclampsia (LOPE), starting after 34 weeks (80% of all cases) [36,37].

The variability in symptom severity, the effectiveness of preventive treatments, and long-term outcomes differ between the two subtypes as the result of different pathogenic mechanisms [38]. EOPE’s etiology has been directly linked to early defects in placental development, leading to reduced uteroplacental perfusion, hypoxia, and oxidative stress [39].

Many studies demonstrated that an adverse uterine environment leads to an increased risk for IUGR, low birth weight, neonatal death, congenital heart defects, and more [37,40,41,42].

On the other hand, LOPE is identified as “maternal preeclampsia” due to maternal endothelial dysfunction [43]: this condition leads to an imbalance between angiogenic and anti-angiogenic proteins [44] and it is related to a later maternal predisposition to metabolic disorders, such as dyslipidemia, long-term diabetes [13,38], and cardiovascular disorders, like essential hypertension and coronary heart disease [45,46]. In a reciprocal way, environmental factors like chronic hypertension, obesity, and diabetes significantly increase the likelihood of developing preeclampsia [47,48].

Placental growth factor (PIGF), a pro-angiogenic molecule, and soluble FMS-like tyrosine kinase-1 (sFlt1), an anti-angiogenic factor, are two of the most-studied biomarkers involved in PE [49,50,51]. Low levels of circulating PIGF and high levels of sFlt-1 are normally found in patients with PE [52,53], leading to an elevated ratio, recently studied as a predictive biomarker of PE [54].

Moreover, there are a few more candidate biomarkers for PE prediction: first-trimester pregnancy-associated plasma protein A (PAPPA-A) [55], Pentraxin 3 [56], microRNAs [57] and some circulating HDP-associated peptides (P-2081, P-2091, P-2127, P-2209, P-2378, P-2858, and P-3156) [58,59].

### 3.1. Placental Dysfunction and the “Two-Stage Model”

Placental dysfunction, and, in particular, the “two-stage theory”, has been one of the most studied hypotheses behind PE, although the comprehension of PE pathogenesis is still limited. Here, we report an overview of the “two-stage theory”:Stage 1 (placental stage): The initial stage involves poor placentation, where the extravillous trophoblasts fail to properly invade and remodel maternal spiral arteries [60]. This vascular transformation is often incomplete or absent, particularly in the junctional zone of the myometrium, in women who subsequently develop PE. Consequently, this results in inadequate blood flow to the placenta and ischemia, triggering the release of anti-angiogenic factors such as sFlt-1 and PlGF [61,62] but also of immune mediators and ROS [47,63].Stage 2 (maternal symptoms): These placental factors enter the maternal bloodstream, leading to systemic endothelial dysfunction and the onset of clinical symptoms, such as hypertension and proteinuria [64,65].

### 3.2. Angiogenic Imbalance

One of the hallmarks of preeclampsia is an imbalance between pro-angiogenic and anti-angiogenic factors. In healthy pregnancies, pro-angiogenic factors like VEGF and PlGF promote blood vessel formation and placental growth. On the other hand, PlGF levels are significantly reduced in PE, while anti-angiogenic factors such as sFlt-1 and soluble endoglin (sEng) are elevated. This imbalance disrupts normal vascular function, contributing to widespread endothelial damage and the development of systemic hypertension [64,66]. The sFlt-1/PlGF ratio is increasingly popular in clinical settings to predict the onset and severity of preeclampsia [48,63,67]. Furthermore, therapeutic strategies targeting this imbalance, such as the use of VEGF agonists or neutralizing antibodies against sFlt-1, are being explored as potential treatments [47].

### 3.3. Endothelial Dysfunction and Systemic Oxidative Stress

Another pathogenetic hypothesis involves endothelial dysfunction. The endothelium is damaged by the combination of oxidative stress, inflammation, and anti-angiogenic factors [60]. This results in impaired vascular tone regulation [63,66], contributing to the development of a PE-associated prothrombotic state [68].

### 3.4. Genetic and Epigenetic Aspects

The genetic side has been investigated to assess whether there is a predisposition to PE. Variants in genes related to angiogenesis, immune regulation, and placental development have been associated with an increased risk of the disorder [64]. Among them, polymorphisms in the *FLT1* gene (which encodes sFlt-1) are linked to higher levels of sFlt-1 and a higher risk of developing PE [69].

Moreover, epigenetic modifications such as altered microRNA (miRNA) expression also influence transcriptional activity related to trophoblast invasion and vascular function [70]. For example, chromosome 19 microRNA cluster (C19MC) is supposed to be heavily involved in PE. Dysregulation of miR-519d, miR-517-5p, and others from this cluster affects angiogenesis: miR-519d suppresses trophoblast invasion, while miR-517-5p impairs syncytiotrophoblast differentiation, leading to a decreased ability of cells to invade and proliferate [70,71].

Also, several long non-coding RNAs (lncRNAs) have been associated with PE. For instance, IGFBP1 is linked to placental insufficiency, while EGFR-AS1 influences cell proliferation by modulating the JAK-STAT signaling pathway [72,73].

Lastly, circular RNA (circRNA) has been suggested to participate in the progression of EOPE. CircRNA_06354 was recently identified by Zhang et al. as playing a role in angiogenesis via the hsa-miR-92a-3p/VEGF-A pathway; therefore, a dysregulated expression impairs trophoblast invasion into spiral arteries, contributing to EOPE [74].

These aspects highlight a complex interaction between genetic and environmental factors in determining the pathogenetic mechanisms underlying PE [64].

## 4. Immune System and Immune Dysfunctions in Preeclampsia

Pregnancy can physiologically develop when proper immune tolerance between mother and fetus occurs, in order to prevent fetal rejection [51,75]. Natural killer cells (NK cells) and T cells located in the decidua play a crucial role in the maternal immune tolerance to the fetus through well-known strategies of antigen presentation, inhibitory cytokine production, and cytolysis of targeted cells [76,77,78]. The conflict between maternal and paternal genes leads to higher NK cell activity, decreased regulatory T-cell (Treg) number, and other changes in immune mediators’ production [79]. The dysregulation of all these mechanisms prevents the establishment of a pregnancy-favorable environment and leads to various pregnancy disorders, such as preeclampsia and recurrent spontaneous abortion [78,80].

Indeed, according to recent studies, an abnormal shift to T helper (Th1) cell phenotype, higher doses of pro-inflammatory molecules and higher levels of NK cells were found in PE patients [76,81]. This inflammatory environment supports the production of pro-inflammatory cytokines, autoantibodies, and ROS, with a worsening effect on the ongoing ischemic process, as previously described [51]. A schematic representation of these changes is shown in Figure 1.

The inflammatory response to oxidative stress is self-perpetuating and leads to a chronic condition fundamental for PE pathogenesis. This compromised self-regulation process resembles the immune dysfunctions in autoimmunity and is also involved in some other pregnancy complications, such as gestational diabetes mellitus (GDM) and pre-term delivery [13]. The existing connection between pregnancy and autoimmune diseases is extremely complex: in patients affected by autoimmune diseases, the abnormal activation of maternal innate and adaptive immune systems can interfere with placentation and tolerance of the fetal semi-allograft, with a higher risk of PE, IUGR, prematurity, congenital heart block, and autoimmune diseases in the offspring and even stillbirth [14,82,83,84,85]. It is well known that systemic lupus erythematosus (SLE), antiphospholipid syndrome (APS) (mostly the active disease), and hypothyroidism are related to an increased risk of PE [86,87,88]. Furthermore, PE patients showed higher levels of antibodies against angiotensin II type 1 receptor (AT1-AAs) [89]: these antibodies show a persistent effect on vasoconstriction and lead to endothelial damage and insufficient trophoblast invasion, typical conditions of PE pathogenesis [90]. Furthermore, these autoantibodies could increase sFlt-1 secretion, but it is still uncertain whether these anomalies are pathogenic or epiphenomena of PE [91]. Finally, prior exposure to paternal/fetal antigens could have a protective role for PE: the maternal immune system’s progressive exposure to paternal semen leads to a persistent tolerance toward the fetal semi-allograft until pregnancy term [2,92,93]. For this reason, long inter-pregnancy interval (>10 years), pregnancies with different paternity, and nulliparity could represent possible PE risk factors [93,94,95]. It has also been noticed that PE incidence is higher in pregnancy conceived via intra-cytoplasmic sperm injection (ICSI) [96,97,98] or with an egg and/or sperm donor [95,99].

## 5. Gut Microbiota and Preeclampsia

In consideration of the increasing evidence of the microbiota’s impact on immune regulation, the hypothesis on the possible effect of intestinal dysbiosis and consequent altered immune modulation on PE development could be an interesting and new point of view. Indeed, this represents a very recent field of research, and the current literature on this topic is constantly growing. We found 28 relevant reports assessing the differences among PE patients and controls of gut flora composition, including bacteria, viruses, and fungi (i.e., mycobiome) (Table 1) and its metabolites, and lastly the possible causative role of the microbiota, as described in the following paragraphs.

### 5.1. Differences in Gut Microbiota Composition in PE Patients

One of the first studies evaluating microbial composition in PE patients was conducted by Liu et al., who demonstrated in PE patients an overall increase in pathogenic bacteria, *Clostridium perfringens* and *Bulleidia moorei*, and a reduction in propionic-producing bacteria, known as probiotic bacteria, such as *Coprococcus catus* [100]. Similar results on PE-induced dysbiosis were found in another cohort of patients, showing an inverse correlation between *Lactobacillus* abundance and SBP, DBP, and proteinuria [101]. Furthermore, a recent pilot study confirmed a lower abundance of *Coprococcus* in HDP patients, while a higher abundance of pathogenic bacteria (*Rothia*, *Actinomyces*, and *Enterococcus*) and *Escherichia* as the dominant bacterium were found in the case group [102]. Other studies demonstrated significant alterations in microbial composition during the third trimester compared to the second trimester in PE patients, with a reduction in *Firmicutes* and an increase in *Bacteroidetes*, *Proteobacteria*, *Blautia,* and *Faecalibacterium* [103,104]. Similarly, Wu et al. evaluated the longitudinal variation of gut microbiota in pregnant patients, and the group who would later develop PE or gestational hypertension was significantly enriched for *Polycyclovorans* and *Pelomonas* at the end of the first trimester and at term of pregnancy [105]. Then, Lv et al. found a significant enrichment of *Blautia*, *Ruminococcus2*, *Bilophila,* and *Fusobacterium* and a significant depletion of *Faecalibacterium*, *Gemmiger*, *Akkermansia*, *Dialister*, and *Methanobrevibacter* in antenatal PE samples; these taxonomic alterations were similar to postpartum samples [106]. Similarly to another study, *Blautia* and *Ruminococcus2* were positively correlated with obesity, dyslipidemia, glycosyltransferases, biotin metabolism, and oxidative phosphorylation pathways [106,107]. Afterwards, the same group found an enrichment of other species belonging to *Pauljensenia* and *Collinsella* in women with early-onset PE through deep shotgun metagenomic sequencing, showing an improvement in the accuracy of species identification with this technique [108]. Different results were obtained by Zhao et al., who found lower levels of *Bacteroidetes*, *Ruminococcus,* and *Oscillospira* and an increasing ratio of *Firmicutes* and *Bacteroidetes* in the PE group compared to controls [109]. Although fungi account for about 0.1% of the gut microbiome, they still represent a substantial part of the microbiome biomass. For this reason, the mycobiome, the fungal component of the gut flora, was also evaluated. Meijer et al. found a similar bacterial microbiome composition in the PE group compared to previous studies, but, analyzing the mycobiome, they found no significant differences in the abundance of *Saccharomycetes*, *Dothideomycetes* and *Sordariomycetes* classes between the case and control groups, except for *ASV-36*, corresponding to the genus *Peniophora* [110]. In conclusion, alpha diversity (i.e., observed richness of taxa of an average sample) is significantly lower in PE patients, while, according to a recent meta-analysis, a comparison of relative abundance of *Bacteroidetes*, *Firmicutes*, *Actinobacteria*, and *Proteobacteria* showed no significant results [111].

### 5.2. The Causative Role of Microbiota in PE Pathogenesis

It is well known that there is a different intestinal microbial composition in patients affected by hypertension. Indeed, several cross-sectional studies in humans found higher abundances of Gram-negative bacteria, including *Klebsiella*, *Parabacteroides*, *Desulfovibrio*, and *Prevotella*, and lower abundances of SCFA-producing bacteria, including *Ruminococcaceae*, *Roseburia*, and *Faecalibacterium* spp. [112]. Similarly, in PE patients, SBP and DBP significantly acted on microbial diversity and structure compared to other clinical parameters such as age, body mass index (BMI), gestational age, fasting glucose, and glycosylated hemoglobin [108]. To understand the causative role of the microbiota in the pathogenesis of this disease, Chen et al. performed a fecal microbiota transplantation (FMT) in an antibiotic-treated mouse model: they showed an increase in SBP, proteinuria, histopathological structural alterations in placenta, Th17, and higher intestinal permeability in mice transplanted with PE microbiota [25]. Similar data were found in other studies, with an increase in peripheral Th17 cells, increased serum levels of tumor necrosis factor α (TNF-α), interleukin 17 (IL-17), IL-1β, IL-6, and LPS in PE patients and, similarly, in a preeclamptic mouse model after FMT [28,109,113]. Lastly, four different studies used mendelian randomization (MR) to understand the effect of microbial changes on PE. Three of these studies showed a protective role against HDPs and PE of *Bifidobacterium*, *Collinsella*, *Enterorhabdus*, *Eubacterium*, *Lachnospiraceae*, *Tyzzerella 3*, and *Methanobrevibacter*, while *LachnospiraceaeUCG010*, *Olsenella*, *RuminococcaceaeUCG009*, and *Ruminococcus2* were related to a significantly increased risk for gestational hypertension [105,114,115]. Similarly, another study showed a potential protective role of *Streptococcus* by affecting TNF-α, a protective effect of the *Enterobacteriales* order, and a pathological role of *Olsenella*, through modulation of unsaturated fatty acid biosynthesis with secondary effects on vasoconstriction, oxidative stress, inflammation, and coagulation [28].

### 5.3. Metabolomic Analysis of Intestinal Microbiota PE Patients

There is increasing evidence on the effects of microbial metabolites on the immune and endocrine systems, and these variations may be associated with obesity, elevated blood pressure, and other comorbidities [116]. Recently, Cui et al. reviewed seven human cohort-controlled trials, six of them comparing gut microbiota composition and its association with SCFAs between healthy pregnant women and patients affected by PE [117]. These studies showed that SCFAs are strictly related to gut microbiota abundance; specifically, serum acetate levels seemed to be protective against PE. Moreover, PE patients had significantly decreased serum acetate, propionate, and butyrate concentrations, related to the concomitant reduction in *Coprococcus* [20,118,119]. It was recently demonstrated that *Firmicutes* had a lower abundance in PE patients, and SCFAs produced by this phylum may influence blood pressure and renal function [120]. Another recent study evaluated the variation of SCFAs in serum samples of 38 PE patients and 29 healthy pregnant women after 28 gestational weeks: significantly increased levels of acetic acid, propionic acid, isobutyric acid, and valeric acid were found in the PE group, and these metabolites were positively correlated to blood pressure and urea levels and negatively correlated with the gestational week and fetal birth weight [121]. Additionally, according to the results of ROC curve analyses, the authors demonstrated the possible use of these SCFAs as serum biomarkers for the diagnosis of PE, especially isobutyric acid, which showed the highest sensibility and specificity [121]. On the other hand, bioactive metabolites derived from the gut microbiota, such as trimethylamine N-oxide (TMAO), were found to be related to higher levels of carnitine and choline, sterol and cholesterol metabolism, and atherosclerotic-like lesions in the spiral arteries in placentas obtained from PE patients [122,123]. Furthermore, lower plasmatic levels of betaine and higher plasmatic levels of choline were associated with higher HDP risk in the early second trimester of pregnancy [124].

**Table 1 ijms-25-10099-t001:** The intestinal microbial composition in patients affected by preeclampsia (PE) according to the most recent literature.

Author, Year	Country	Study Design	Methods	Number and Type of Patients	Number of Stool Samples	Phylum Levels	Clinical Impact on Maternal and Neonatal Health
Liu et al., 2017 [100]	China	Case–control	16S rDNA gene amplicon sequencing	26 PE patients (11 mild symptoms and 15 severe symptoms) vs. 3 controls	100	Similar composition in the dominant phyla Increased level of *Cyanbacteria* in PE group	N/A
Lv et al., 2019 [106]	China	Case–control	16S rRNA gene amplicon sequencing	78 PE patients vs. 72 controls	Antepartum: 48 PE vs. 51 controls; 1 week postpartum: 35 PE vs. 17 controls; 6 weeks postpartum: 18 PE vs. 11 controls	Similar composition in the dominant phyla	Positive correlation between DBP and SBP levels and *Anaerococcus*, *Ruminococcus2*, *Fusobacterium*, and *Oribacterium*, positive correlation among birth weight and PE-depleted genera.
Wang et al., 2020 [103]	China	Nested case–control	16S rRNA gene amplicon sequencing	25 PE patients vs. 25 controls	20–24 weeks gestation: 25 PE vs. 25 controls; 32–34 weeks gestation: 15 PE vs. 22 controls	In T2, no significant differences between PE group and controls; in T3, significantly lower levels of *Firmicutes*, significantly higher levels of *Bacteroidetes* and *Proteobacteria* in PE group vs. controls	N/A
Huang et al., 2021 [101]	China	Case–control	16S rRNA gene amplicon sequencing	26 PE patients, 25 APG ^§^ patients vs. 28 controls and 21 healthy non-pregnant patients	N/A	Similar composition in the dominant phyla Significantly increased percentage of *TM7* in the APG group	Significant correlation of *g_1_68*, *g_Porphyromonas*, *g_Mobiluncus*, and *g_Lactococcus* with SBP and DBP. Significant inverse correlation among *g_Lactobacillus* and SBP, DBP and UP.
Miao et al., 2021 [107]	China	Case–control	16S rRNA gene amplicon sequencing	12 PE patients vs. 8 controls	20	Slightly increased levels of *Firmicutes* and significantly decreased levels of *Actinobacteria* in PE group	Positive correlation between *Blautia* and maternal age, pregestational weight, HCT, PCR, TGs, and LDL. Positive correlation between *Ruminococcus* and pregestational weight, pregestational BMI, antepartum weight, antepartum BMI, LBP, and TGs. Negative correlation between *Bifidobacterium* and SBP, DBP, cholesterol, AST, and TGs.
Lv et al., 2022 [108]	China	Case–control	Shotgun metagenomic sequencing	40 PE patients vs. 37 controls	77	Major phyla included *Firmicutes*, *Actinobacteria*, and *Bacteroidetes* in both groups	N/A
Geldenhuys et al., 2022 [104]	South Africa	Case–control	16S rRNA gene amplicon sequencing	10 PE patients vs. 11 controls	6 PE vs. 8 controls	No significant difference between two groups	N/A
Wu et al., 2023 [105]	China	Nested case–control	16S rRNA gene amplicon sequencing	29 GH patients and 17 PE patients vs. 30 controls	17 PE and 29 GH vs. 30 controls	No significant difference between GH, PE, and control groups during the early, middle, and late trimester	In the early trimester, positive correlation among *Polycyclovorans* and DBP. In the middle trimester, negative correlation between *Acidibacter*, *Amphiplicatus*, and *Shewanella* and DBP. In the late trimester, *Pelomonas* positively correlated with SBP, *Thermomonas* negatively correlated with DBP.
Wu et al., 2023 [102]	China	Case–control	16S rRNA gene amplicon sequencing	14 PE patients vs. 9 controls	N/A	Detection of *Euryarchaeota* only in controls	N/A
Zhao et al., 2023 [109]	China	Case–control	16S rRNA gene amplicon sequencing	41 PE patients vs. 45 controls	86	Significantly lower relative abundance of *Bacteroidetes* and significantly higher relative abundance of *Proteobacteria* in PE group	N/A

Abbreviations: rRNA: ribosomal RNA; PE: preeclampsia; N/A: not available; T2/T3: second/third trimester; APG: abnormal placental growth; GH: gestational hypertension; DBP: diastolic blood pressure; SBP: systolic blood pressure; UP: urinary protein concentration; HCT: hematocrit; PCR: C-reactive protein; TGs: triglycerides; LDL: low-density lipoprotein cholesterol: LBP: lipopolysaccharide-binding protein; BMI: body mass index; AST: aspartate aminotransferase. ^§^ Abnormal placental growth diagnosis: abnormal PIGF Multiple of the Median (MoM) values in early and middle pregnancy.

## 6. Discussion

The intestinal microbiota is a highly complex system involved in various metabolic and immunological functions. It changes dynamically from childhood to adulthood, including during motherhood. During pregnancy, the gut microbiota shifts toward the end of gestation, with an increase in Proteobacteria and Actinobacteria, and a decrease in SCFAs. These changes appear to be related to physiological processes such as weight gain, low-grade inflammation, and insulin resistance [125]. The intestinal microbiota can also be influenced by factors like medication (e.g., antibiotics), diet, physical activity, and lifestyle, leading to the high heterogeneity of studies evaluating the microbiota in pregnant women. In this review of the most recent literature, preeclampsia (PE) was associated with intestinal dysbiosis compared to healthy controls. However, larger sample sizes and more standardized study protocols are needed to reduce the influence of confounding factors. Due to these limitations, a specific combination of microbes related to this disease has not been consistently reported in studies. Some of the selected studies evaluated the impact of these microbial changes on clinical parameters used for diagnosing preeclampsia [101,105,106,107], while others examined the correlation between gut dysbiosis and inflammatory or intestinal permeability markers [25,28,109,113]. Patients with PE showed higher plasmatic levels of LPS, elevated levels of pro-inflammatory Th17 cells, and increased pro-inflammatory cytokines. In a mouse model, LPS was involved in key pathways for blood pressure regulation, leading to decreased baroreflex sensitivity, increased Toll-like receptor (TLR) and TNF-alpha expression in the paraventricular nucleus, elevated heart rate, and increased norepinephrine levels [126]. Additionally, mice transplanted with PE microbiota exhibited lower expression of Claudin-4 and Occludin genes, and a marked reduction in zonula occludens-1 and -2, Claudin-4, and Occludin proteins in the colon [25]. These proteins form intestinal tight junctions, and zonulin, a protein secreted by the basal lamina of the intestinal epithelium, activates intracellular signaling pathways involved in paracellular permeability. Zonulin is one of the most-studied markers of intestinal permeability [127]. Increased gut permeability is linked to systemic translocation of metabolites and endotoxins, potentially amplifying intestinal permeability. Microbiota dysbiosis can induce low-grade chronic inflammation, impairing intercellular connections by downregulating tight junctions, and causing a “leaky gut” [128], a condition associated with increased endometrial inflammation in patients suffering from recurrent pregnancy loss [129]. According to the selected studies, preeclamptic patients exhibited gut microbial shifts, higher levels of inflammatory markers, and increased intestinal permeability, leading to the hypothesis that a leaky gut could be a potential pathological pathway involved in the development of preeclampsia. However, the current evidence is insufficient to establish a cause–effect relationship, and further studies are required to confirm this hypothesis. Lastly, several studies have annotated the functional characteristics of microbiota in PE using the KEGG (Kyoto Encyclopedia of Genes and Genomes) database [130]. Enzymes involved in transport systems (e.g., bacitracin/lantibiotics, rhamnose, and maltooligosaccharide transport), signal transduction, environmental adaptation, and cell motility were significantly upregulated in the PE group, suggesting that these mechanisms could contribute to intestinal dysbiosis [102]. From metabolomic analyses, the most consistent evidence identified isobutyric acid and trimethylamine N-oxide (TMAO) as early predictors of PE [121,123]. Trimethylamine (TMA) is mainly produced by the families Clostridia and Enterobacteriaceae, with the degradation of nutrients commonly found in dietary products like meat and eggs, and then it is oxidized by the hepatic enzymes. TMAO has various effects on inflammation (e.g., increasing TNF-α and IL-1B, decreasing IL-10), cholesterol metabolism, and thrombosis (e.g., inducing platelet hyperreactivity) [112]. These findings have been raised from few studies, so it is necessary to develop further associations between these metabolites and preeclampsia to introduce them into clinical practice.

## 7. Conclusions

The analysis of the intestinal microbiota in preeclampsia is a growing area of research. Over the past few decades, studies have increasingly demonstrated a connection between gut dysbiosis and this disease. However, it is essential to establish a microbial baseline in healthy pregnant women to allow for better comparisons between cases and controls and to reduce the heterogeneity of studies. Understanding the immunological impact of microbiota in preeclampsia could be a crucial step toward a more comprehensive study of its pathogenesis. Randomized clinical trials are necessary to provide stronger evidence on the relationship between microbiota and preeclampsia and to identify metabolic features as early diagnostic markers for PE. In this way, earlier preventive strategies could be developed to reduce the disease’s impact on maternal and fetal outcomes.

## Figures and Tables

**Figure 1 ijms-25-10099-f001:**
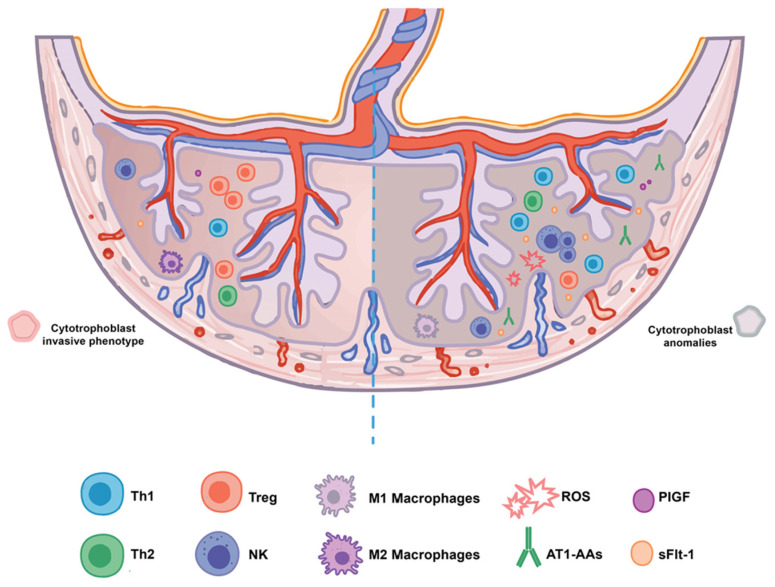
Defective trophoblastic phenotype and related immunological dysfunctions in preeclampsia. On the left side, the physiological placentation processes are represented: the cytotrophoblast cells acquire an invasive phenotype to enhance efficient vascular anastomoses, and decidual T cells, NK cells, and M2 macrophages establish the maternal immune tolerance to the fetus. On the right side, there is a schematic representation of immunological changes in preeclampsia. The trophoblast without the invasive phenotype leads to the abnormal development of narrow vessels and endothelial damage (also due to higher levels of AT1-AAs) and increased ROS production. These changes cause an imbalance of angiogenic and anti-angiogenic factors (PIGF and sFlt-1) and a pro-inflammatory environment (Th1 shift, increasing levels of NK and M1 macrophages). Acronyms: Th1/Th2: T helper 1/2 cells; Treg: regulatory T cells; NK: natural killer cells; ROS: reactive oxygen species; AT1-AAs: antibodies against the angiotensin II type 1 receptor; PIGF: placental growth factor; sFlt-1: soluble FMS-like tyrosine kinase-1.

## Data Availability

No new data were created.

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
