# Peer review of "The Hidden Relationship between Intestinal Microbiota and Immunological Modifications in Preeclampsia Pathogenesis"

_ijms, 2024, doi:10.3390/ijms251810099_

Round 1

Reviewer 1 Report

Comments and Suggestions for Authors

 Thank you very much for allowing me to review the review article titled “The hidden relationship between intestinal microbiota and immunological modifications in preeclampsia pathogenesis “(ijms-3184866). The aim of this narrative review is to investigate the role of the immune system in the pathogenesis of preeclampsia and to summarize the most recent literature on the potential link between preeclampsia and alterations in the intestinal microbiota.

The abstract is a crucial part of the work, as it serves as the introduction and should therefore contain as much comprehensive information as possible to give an overview of the entire study. Consequently, the abstract should not only address the significance of the topic and the objective of the review but also include details such as the methodology or design used, the time period covered, the database consulted, the inclusion criteria, the number of articles reviewed, the main findings, and the conclusions in a concise manner to provide a complete overview.

The introduction discusses the importance of preeclampsia and its significant consequences for both the mother and the baby. It then introduces the relevance of the intestinal microbiota, particularly between lines 61 and 67. This discussion, however, should be expanded, as it justifies the need for this review. For example, further elaboration on the pathophysiological basis of this relationship or the current knowledge on the topic would be beneficial.

Section four, which addresses the intestinal microbiota and immune dysfunction, presents various studies, most of which are animal-based and not directly related to preeclampsia. It is unclear how this section justifies its inclusion in the review.

Section five does not adequately highlight the importance of the intestinal microbiota in immune regulation, only touching on the potential link with the pathogenesis of preeclampsia in section 52, despite most of the information referring to other comorbidities.

In Table 1, at the very least, the study designs utilised should be indicated, along with information on the impact on maternal and neonatal health, if such differences are reported in the studies.

The conclusion presented in the manuscript is not a true conclusion but rather a summary of what has been discussed. This section needs to be rewritten, as the conclusion should reflect the review's contribution to the existing knowledge.

Overall, I believe this work would benefit from including a minimum description of the methodology. Reviews are highly valued as they allow us to integrate knowledge on a topic from recent years into a single work. However, this poses a challenge in identifying different reviews conducted within the same period or connecting past reviews with future ones. Therefore, the study would benefit from including, at the very least, the type of review conducted, which appears to be a comprehensive review, the time period covered, the databases used, the inclusion and exclusion criteria applied, and the number of articles reviewed. It would also be highly beneficial to include a discussion section that addresses the strengths and weaknesses of current knowledge and assesses this review in terms of identifying key areas for future research.

Moreover, this review should focus strictly on the topic presented in the title and objective, removing any aspects not directly related to the microbiota and preeclampsia, as they fall outside the scope of this work.

Author Response

Comment 1: The abstract is a crucial part of the work, as it serves as the introduction and should therefore contain as much comprehensive information as possible to give an overview of the entire study. Consequently, the abstract should not only address the significance of the topic and the objective of the review but also include details such as the methodology or design used, the time period covered, the database consulted, the inclusion criteria, the number of articles reviewed, the main findings, and the conclusions in a concise manner to provide a complete overview.

Response 1: Thank you for your valuable suggestions. We have revised the abstract (lines 12-24) by summarizing the background section. Additionally, we have highlighted the methodology, including the databases consulted and the inclusion criteria for selected articles, as well as the main results of our review, such as the number of evidence and key findings, per your recommendations.

Comment 2: The introduction discusses the importance of preeclampsia and its significant consequences for both the mother and the baby. It then introduces the relevance of the intestinal microbiota, particularly between lines 61 and 67. This discussion, however, should be expanded, as it justifies the need for this review. For example, further elaboration on the pathophysiological basis of this relationship or the current knowledge on the topic would be beneficial.

Response 2: Thank you for the advice. We cited the importance of gut microbiota in physiological processes occurring during pregnancy, specifically for the fine immune regulation (line 67-83). We have also included the relationship between microbiota and metabolic effects in obese pregnant women, which is further analyzed in subsequent sections (line 89-95).

Comment 3: Section four, which addresses the intestinal microbiota and immune dysfunction, presents various studies, most of which are animal-based and not directly related to preeclampsia. It is unclear how this section justifies its inclusion in the review.

Response 3: In agreement with your suggestion, we have deleted section four ("Intestinal microbiota and immune dysfunction"). Instead, we have integrated the discussion on the immune system's interaction with microbiota into the introduction (lines 67-95), as advised in your previous comments.

Comment 4: Section five does not adequately highlight the importance of the intestinal microbiota in immune regulation, only touching on the potential link with the pathogenesis of preeclampsia in section 52, despite most of the information referring to other comorbidities.

Response 4: The objectives in section five (line 268-368), titled "Gut Microbiota and Preeclampsia," were as follows:

  1. To evaluate dysbiosis in patients with preeclampsia compared to controls.
  2. To assess metabolomic analyses conducted in patients with preeclampsia.
  3. To highlight the impact of microbial dysbiosis on clinical parameters used for preeclampsia diagnosis (e.g., blood pressure, urine protein concentration, etc.), as mentioned in the introduction and reflected in Table 1.

The evidence regarding the microbiota's potential role in immune regulation in preeclampsia is limited, as discussed in section 5.2 (line 403-427), which we acknowledge as a limitation of our review in the discussion section.

Comment 5: In Table 1, at the very least, the study designs utilised should be indicated, along with information on the impact on maternal and neonatal health, if such differences are reported in the studies.

Response 5: We have also modified the structure of Table 1 (line 380-389) based on your feedback, including details on study design, methods for microbiota detection, and the clinical impact of microbial changes related to preeclampsia on maternal and neonatal outcomes.

Comment 6: The conclusion presented in the manuscript is not a true conclusion but rather a summary of what has been discussed. This section needs to be rewritten, as the conclusion should reflect the review's contribution to the existing knowledge.

Response 6: We have rewritten the conclusion (line 443-453), emphasizing the main findings and future research prospects, as you suggested. 

Comment 7: Overall, I believe this work would benefit from including a minimum description of the methodology. Reviews are highly valued as they allow us to integrate knowledge on a topic from recent years into a single work. However, this poses a challenge in identifying different reviews conducted within the same period or connecting past reviews with future ones. Therefore, the study would benefit from including, at the very least, the type of review conducted, which appears to be a comprehensive review, the time period covered, the databases used, the inclusion and exclusion criteria applied, and the number of articles reviewed. It would also be highly beneficial to include a discussion section that addresses the strengths and weaknesses of current knowledge and assesses this review in terms of identifying key areas for future research.

Response 7: Thank you once again for your insightful suggestions. We have changed the structure of the manuscript to include a clear division between the introduction (line 29-99), methods (lines 101-112), main results (lines 273-276), and discussion (lines 390-442), addressing all points raised in your comments.

Best Regards

Reviewer 2 Report

Comments and Suggestions for Authors

Dear authors,

Congratulations on this very interesting work. It is a topical article. 

I suggest mentioning in the introduction the classification of preeclampsia according to ISSHP.

I would like to mention the definition of fetal growth restriction (according to ISUOG). 

Best regards,

Author Response

Comment 1: I suggest mentioning in the introduction the classification of preeclampsia according to ISSHP. I would like to mention the definition of fetal growth restriction (according to ISUOG). 

Response 1: Dear Reviewer, thank you for your support. We inserted the criteria ISSHP (line 46-50) and ISUOG (line32-35) for preeclampsia and fetal growth restriction respectively, as you suggested.

Best Regards

Reviewer 3 Report

Comments and Suggestions for Authors

 This work is an excellent review of the relationship between intestinal microbiota and the preeclampsia pathogenesis. The authors first talk about the problem of hypertensive disorders of pregnancy. They gave statistics on this health problem. Next, they describe preeclampsia as one of the most important causes of maternal death and explain that preeclampsia has been associated with chronic diseases in mothers who are still alive. Then, the authors mention the different types of preeclampsia and describe its characteristics. Posteriorly, they link the immune system with preeclampsia and discuss how it works in this phenomenon. Subsequently, the authors cite some evidence about the influence of the gut microbiota on immune dysfunction. Last, the authors are concerned about the gut microbiota with preeclampsia and cite some works demonstrating that a specific type of microbiota could be responsible for preeclampsia disorder. I consider this a great review.

Author Response

Dear Reviewer, thank you for you support. 

Best Regards

Round 2

Reviewer 1 Report

Comments and Suggestions for Authors

I have reviewed the article titled " ijms-3184866_ The hidden relationship between intestinal microbiota and immunological modifications in preeclampsia pathogenesis ", as well as the authors' response to the suggestion made to enhance the clarity of the work. The authors have indeed made a considerable effort to improve the quality of this paper, but there are still significant gaps that need to be addressed, which I will outline below.

In the abstract, the time period covered in the literature review is not specified, which is essential for connecting this review with previous and future studies. In a research paper, it is not appropriate to state "approximately 30 studies"—the exact number of studies included in the review must be specified. Additionally, the abstract should include a minimum of key findings and conclusions.

The end of the introduction should clearly and precisely state the objective of this review. I noticed a discrepancy between the manuscript, which simply mentions the objective as conducting a review, and the authors' response, which outlines three objectives as if this were an original research study. The objective should be clearly defined, keeping in mind that this is a narrative review. This will help maintain coherence throughout the paper.

In the abstract, the authors describe the work as a narrative review; however, in the methodology section, they refer to it as a comprehensive review. Please clarify this, as they are different types of reviews.

In line 102, the authors mention using various combinations of keywords. As this is a scientific paper, it must be precise—please specify the exact keywords used so that the study can be replicated. In line 105, they state that recent evidence was also considered, which is quite vague—please specify. Lines 105–111 mention "designs"; this should be clarified as inclusion criteria, which means that both inclusion and exclusion criteria are currently missing. In line 110, the phrase "more than 200 articles" is again imprecise—please be specific, as this is a scientific article.

Author Response

I have reviewed the article titled " ijms-3184866_ The hidden relationship between intestinal microbiota and immunological modifications in preeclampsia pathogenesis ", as well as the authors' response to the suggestion made to enhance the clarity of the work. The authors have indeed made a considerable effort to improve the quality of this paper, but there are still significant gaps that need to be addressed, which I will outline below.

In the abstract, the time period covered in the literature review is not specified, which is essential for connecting this review with previous and future studies. In a research paper, it is not appropriate to state "approximately 30 studies"—the exact number of studies included in the review must be specified. Additionally, the abstract should include a minimum of key findings and conclusions.

We included the time period of the review, the exact number of papers included analyzing the microbial composition in preeclamptic patients, and the key findings - intestinal dysbiosis and higher inflammatory levels in preeclampsia-, as you suggested (line 11-26).

The end of the introduction should clearly and precisely state the objective of this review. I noticed a discrepancy between the manuscript, which simply mentions the objective as conducting a review, and the authors' response, which outlines three objectives as if this were an original research study. The objective should be clearly defined, keeping in mind that this is a narrative review. This will help maintain coherence throughout the paper.

We modified the end of the introduction in line 98-101, underling that this manuscript is a narrative review.
We mitigate the objectives of our review from our last comment:

“The objectives in section five (line 268-368), titled "Gut Microbiota and Preeclampsia," were as follows: to describe the potential association between dysbiosis and preeclampsia compared to controls, the role of metabolomic analyses, and the impact of microbial dysbiosis on clinical parameters used for preeclampsia diagnosis (e.g., blood pressure, urine protein concentration, etc.), as mentioned in the introduction and reflected in Table 1.

The evidence regarding the microbiota's potential role in immune regulation in preeclampsia is limited, as discussed in section 5.2 (line 403-427), which we acknowledge as a limitation of our review in the discussion section.”

In the abstract, the authors describe the work as a narrative review; however, in the methodology section, they refer to it as a comprehensive review. Please clarify this, as they are different types of reviews.

We remove “comprehensive” term in line 103, as this is a narrative review.

In line 102, the authors mention using various combinations of keywords. As this is a scientific paper, it must be precise—please specify the exact keywords used so that the study can be replicated. In line 105, they state that recent evidence was also considered, which is quite vague—please specify. Lines 105–111 mention "designs"; this should be clarified as inclusion criteria, which means that both inclusion and exclusion criteria are currently missing. In line 110, the phrase "more than 200 articles" is again imprecise—please be specific, as this is a scientific article.

We specified the used keywords in line 103-106.
In agreement with your suggestion, we removed “recent evidence was also considered”.
We clarified the inclusion criteria, and we added the exclusion criteria used to conduct this review (line 107-111). We specified the total number of studies (line 112).

Round 3

Reviewer 1 Report

Comments and Suggestions for Authors

I have carefully reviewed the latest version of the manuscript “ijms-3184866_ The hidden relationship between intestinal microbiota and immunological modifications in preeclampsia pathogenesis”, as well as the authors' response to the suggestions provided to enhance the clarity of their work. The authors have made a significant effort, and I believe it has been worthwhile, as the current version of the article is much more coherent and robust. It presents how the microbiota may be linked to preeclampsia and underscores the need for further research on this topic based on the review conducted.